ecology, evolution, theoretical biology

tipping points, ecosystem resilience, rate-induced regime shifts, eco-evolutionary feedback, environmental change, rate-tipping

**Author for correspondence:**
P. Catalina Chaparro-Pedraza
e-mail: catalina.chaparro@eawag.ch

# Fast environmental change and eco-evolutionary feedbacks can drive regime shifts in ecosystems before tipping points are crossed

P. Catalina Chaparro-Pedraza

Swiss Federal Institute of Aquatic Science and Technology (EAWAG), Dübendorf, Switzerland

(iD) PCC-P, 0000-0001-9801-3878

Anthropogenic environmental changes are altering ecological and evolutionary processes of ecosystems. The possibility that ecosystems can respond abruptly to gradual environmental change when critical thresholds are crossed (i.e. tipping points) and shift to an alternative stable state is a growing concern. Here I show that fast environmental change can trigger regime shifts before environmental stress exceeds a tipping point in evolving ecological systems. The difference in the time scales of coupled ecological and evolutionary processes makes ecosystems sensitive not only to the magnitude of environmental changes, but also to the rate at which changes are imposed. Fast evolutionary change mediated by high trait variation can reduce the sensitivity of ecosystems to the rate of environmental change and prevent the occurrence of rate-induced regime shifts. This suggests that management measures to prevent rate-induced regime shifts should focus on mitigating the effects of environmental change and protecting phenotypic diversity in ecosystems.

## 1. Background

Environmental change is occurring at unprecedented rates [1], and is triggering both ecological and evolutionary responses in ecosystems [2]. Concurrent changes in geographic distribution, population dynamics and phenotypic traits have been documented in multiple wildlife species [3]. These ecological and evolutionary changes together alter the structure and functioning of ecosystems, and therefore the services they provide [4,5], making the need to gain insight into how ecological and evolutionary processes mediate ecosystems' responses to environmental change of the utmost importance.

One of the most concerning effects of environmental change is that ecosystems do not always respond to gradual change in a smooth manner, but that abrupt transitions occur when environmental conditions cross critical thresholds (i.e. tipping points) [6]. These critical transitions, known as regime shifts, have been documented in a variety of ecosystems, including lakes, coral reefs, deserts, woodlands and oceans, and have been attributed to the presence of alternative stable states (ASSs) in ecosystems. There is growing concern over their occurrence because they often alter the availability of ecosystems services, incurring potentially large impacts on society [7]. Hence, this has spurred a large body of research to uncover the mechanisms that trigger regime shifts and develop methods to predict them [8]. So far, the role of ecological processes in the occurrence of regime shifts has been extensively studied, and much attention has been devoted to maintain environmental conditions below safe levels (i.e. below a critical threshold magnitude) to prevent regime shifts (figure 1). Yet, evolutionary processes have been mostly overlooked (but see [9]), potentially limiting our understanding on the occurrence of regime shifts, and thus our ability to prevent them [10].

**Figure 1.** Tipping points, regime shifts and ASSs in ecological theory. According to existing ecological theory, an ecosystem is bistable when two different stable equilibrium states occur for the very same set of environmental conditions, separated by an unstable equilibrium state that marks the border between the basins of attraction of the ASSs. This implies that when environmental conditions are favourable (i.e. low environmental stress) the ecosystem is in the upper branch (state A). If conditions gradually deteriorate, the ecosystem follows the stable equilibrium line until conditions exceed threshold 1. At this point (tipping point TP1), the upper stable equilibrium disappears, and thus a slight increment in environmental stress causes the ecosystem to experience an abrupt regime shift to the lower branch (state B). Most efforts to prevent ecosystem regime shifts therefore focus on maintaining environmental stress below safe levels, in this case, below threshold 1. Once the ecosystem tips, to restore the ecosystem state, it is not sufficient to reduce environmental stress below threshold 1, but to a much lower level of stress indicated by threshold 2 (tipping point TP2). The blue (small vertical) arrows indicate the direction of change when the system is out of equilibrium. (Online version in colour.)

Ecosystem responses to a changing environment are driven by the individual responses of the organisms that are part of it. These responses are in turn driven by phenotypic traits that determine organismal sensitivity to environmental stress, which underlies the response capacity of ecosystems to stress [11]. Phenotypic trait changes have therefore the potential to affect ecosystem responses to environmental change by, for instance, shifting environmental stress thresholds (e.g. tipping points). Furthermore, there is growing evidence that phenotypic trait changes induced by novel selective pressures can influence ecological processes including population dynamics [12], biotic interactions in communities [13] and ecosystem functions such as nutrient cycling and productivity [14]. Given that these ecological processes are central to the biotic feedbacks that maintain ASSs in ecosystems, understanding the effects of environmental change on ecosystems with ASSs requires insight into how ecological, evolutionary and stress dynamics interact.

In this study, I investigate how the interaction between ecological, evolutionary and environmental stress dynamics influences the occurrence of regime shifts in ecosystems with ASSs. Using a general framework to investigate eco-evolutionary dynamics of communities under environmental stress, I incorporate trait evolution in three ecological model systems in which the presence of ASSs has been documented. Subsequently, I examine (1) whether and how trait evolution alters system stability (i.e. asymptotic behaviour), and (2) how ecological, evolutionary and stress dynamics interact and affect the risk of regime shifts (i.e. transient dynamics). Finally, I summarize the findings across the three model systems in the section 'General patterns in eco-evolutionary systems with ASSs under stress'.

## 2. Models and results

I investigate how ecological, evolutionary and environmental stress dynamics influence regime shifts in ecological systems with ASSs in three different model systems corresponding to different ecological scales: a population subjected to an Allee effect; a predator–prey interaction with stage-structured prey; and a shallow lake ecosystem. To do so, I incorporate trait evolution to these systems using standard quantitative genetics techniques [15] and analyse their dynamics using a general multispecies model that couples population dynamics, evolutionary trait dynamics and environmental stress dynamics (a detailed model description can be found in electronic supplementary material, appendix A).

### (a) Population level: demographic Allee effect

A demographic Allee effect occurs when *per capita* growth rate (i.e. fitness) is correlated with population size. Although this effect can influence a wide range of densities, the most commonly described type of Allee effect is that in which *per capita* growth rate decreases with decreasing density [16]. Allee effects have attracted much attention due to their ecological importance as they can increase the probability of population extinction [17], slow the rate of expansion of introduced species [18], and underlie the existence of alternative community states [19]. Given the mathematical simplicity and ecological relevance of demographic Allee effects, I first explore the eco-evolutionary effects of environmental stress on this system.

I consider continuous dynamics of a population with density $N$, which has an Allee effect in the birth process. The *per capita* birth rate depends on the maximum *per capita* birth rate

*b*, the carrying capacity $K$, and the Allee threshold $A$. Below this threshold or above the carrying capacity, the *per capita* birth rate is negative. The *per capita* growth rate (i.e. fitness) results from the difference between the birth rate and death rate of an individual.

To couple the ecological dynamics described above with evolutionary and stress dynamics, I assume the maximum birth rate $b$ to depend on a quantitative phenotypic trait $x$ and environmental variable $E$. An individual's birth rate is maximum (i.e. $b(x, E) = b_{max}$) when its trait value matches the optimal phenotype for the environmental condition $E$, and $b(x, E)$ decreases with increasing difference between its trait $x$ and the optimal phenotype for the environmental condition $E$ (i.e. environmental stress). For instance, the furthest the environmental temperature is from the thermal optimum of an individual, the lowest is its birth rate. The magnitude of the decrease with increasing difference between the actual phenotype and the optimum phenotype is modulated by the parameter $\tau$. Similar trait-based approaches have been used in eco-evolutionary models [20]. I also account for a trade-off between fecundity and survival, which is common in most organisms. The mortality rate is thus the sum of the background mortality $\mu_0$ and an excess mortality $\mu_1$ due to the trade-off. A detailed model description can be found in electronic supplementary material, appendix B.

## (b) Asymptotic stability analysis under fixed levels of environmental stress

I first perform a stability analysis of the system to investigate the equilibrium states at fixed levels of environmental stress $E$ when evolution does not take place due to the absence of genetic trait variance ($\sigma_G^2 = 0$) and when it occurs enabled by genetic trait variance ($\sigma_G^2 > 0$) (figure 2). The presence of genetic trait variance has three effects on the stability behaviour of the system. First, evolutionary trait changes induced by natural selection shift the tipping point that marks the transition to extinction (i.e. collapse threshold, TPc in figure 2) to a higher environmental stress level than when such changes are prevented by the absence of genetic trait variance (e.g. the tipping point in the absence of evolution TPc$_{eco}$ occurs at $E = 1.95$, whereas it occurs at $E = 2.2$ when evolution takes place TPc$_{ecoevo}$; figure 2). Second, in a similar way, evolutionary trait changes shift the tipping point that marks the population invasion threshold (TPi in figure 2) to a higher environmental stress level. As a result, the bistability region is also shifted to higher levels of environmental stress. And third, the basin of attraction of the state corresponding to the existence of the population is larger than that of the system without genetic trait variance (i.e. without evolution) at any level of environmental stress. Following Holling [21], who defines resilience as the size of the basin of attraction of a state, I find that the presence of genetic trait variance enabling evolution increases the resilience of the state in which a population subjected to an Allee effect can persist.

## (c) Transient dynamics when environmental stress increases over time

I also analyse the system dynamics of an evolving population that experiences increasing levels of environmental stress up to a maximum $E_{max}$. I define environmental stress $E = 0$ as the initial condition and simulate gradual increase of

*Proc. R. Soc. B* **288**: 20211192

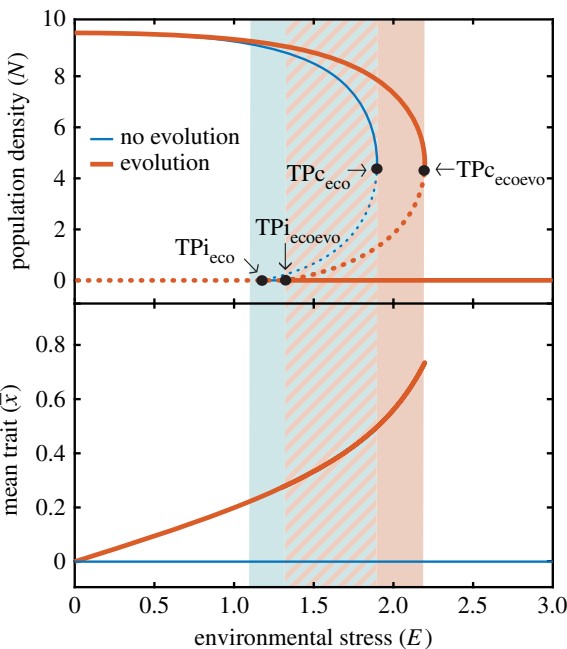

**Figure 2.** Population density (top panel) and mean trait value (bottom panel) in the equilibrium as a function of fixed levels of environmental stress when evolution does not take place (due to absence of genetic trait variance, $\sigma_G^2 = \sigma^2 = 0$; blue lines) and when it does occur ($\sigma_G^2 = \sigma^2 = 0.05$; red lines). Solid lines represent stable equilibrium states and dotted lines represent unstable equilibrium states that separate the basins of attraction of alternative stable equilibrium states in the bistability region. Bistability occurs in the shaded region (blue when there is no evolution, and red when evolution takes place), where the population can persist or go extinct for the very same environmental stress level. Tipping points mark the transition between ASSs (black dots; TPc$_{eco}$, TPc$_{ecoevo}$ correspond to the collapse threshold and TPi$_{eco}$, TPi$_{ecoevo}$ to the invasion threshold). Parameter values: $K = 10$, $A = -1$, $b_{max} = 1$, $\mu_0 = 0.5$, $\mu_1 = 1$, $\tau = 1$. (Online version in colour.)

environmental stress at a constant rate ε. I therefore only study the variation in an environmental condition from this base line, which can take any value in the ecological system (e.g. initial and final temperature might be 20°C and 22°C, respectively; which in the model would be reflected as $E_{max} = 2$). The resulting trajectories were evaluated to determine whether a regime shift that causes the collapse of the evolving population occurs (figure 3). This analysis reveals the existence of three regions with distinct qualitative behaviour in the system:

— The first region corresponds to a population that experiences a gradual increase in environmental stress to a maximum level $E_{max}$ below the tipping point that marks the collapse in the absence of evolution (TPc$_{eco}$). In this region ($E_{max} < 1.95$ in figure 3a), no regime shift can occur, therefore the population always persists.
— A second region corresponds to a population that experiences a gradual increase in environmental stress to a maximum level $E_{max}$ above the tipping point that marks the collapse when evolution is enabled by genetic trait variance (TPc$_{ecoevo}$). In this region ($E_{max} > 2.2$ in figure 3a), a regime shift that causes a population collapse is inevitable. Hence, gradual environmental change driving stress above this threshold (TPc$_{ecoevo}$) always results in population extinction.

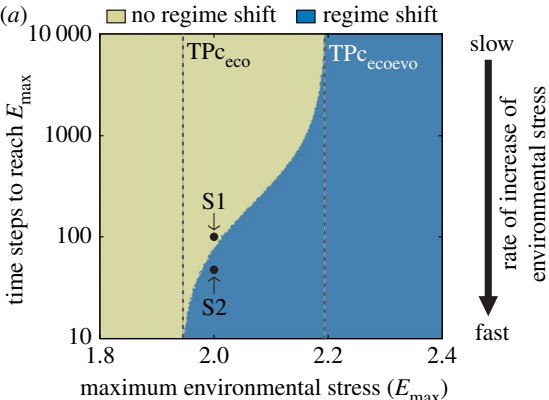

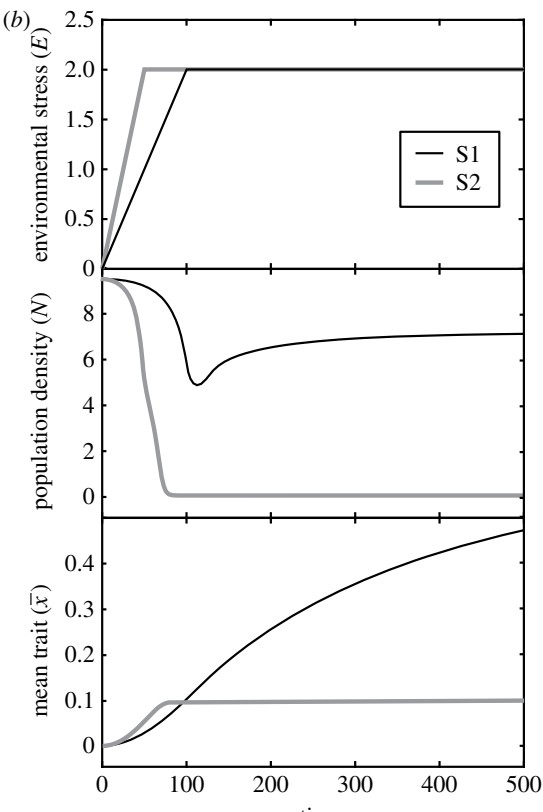

**Figure 3.** (*a*) Occurrence of a regime shift in an evolving population subjected to Allee effect ($\sigma_G^2 = \sigma^2 = 0.05$) as a function of the maximum environmental stress and the rate of the increase in environmental stress (i.e. time steps—days—required to reach the maximum environmental stress level). Vertical dashed lines indicate the values of the tipping points that mark the collapse threshold in figure 2 (TPc$_\text{eco}$, TPc$_\text{ecoevo}$). (*b*) Environmental stress (top panel), population density (middle panel) and mean phenotypic trait (bottom panel) dynamics in the simulations indicated by the points S1 and S2 in (*a*). Parameter values as in figure 2. Additional parameters in (*b*): $E_\text{max} = 2$, $\varepsilon = 0.02$ in S1, and $\varepsilon = 0.04$ in S2 (S1 and S2 only differ in the rate of the increase in environmental stress $\varepsilon$). (Online version in colour.)

— Lastly, there is a transition region corresponding to a population that experiences a gradual increase in environmental stress to a maximum level $E_\text{max}$ above the tipping point that marks the collapse in the absence of evolution (TPc$_\text{eco}$) and below the tipping point when evolution is enabled by genetic trait variance (TPc$_\text{ecoevo}$). In this region ($1.95 < E_\text{max} < 2.2$ in figure 3*a*), both cases (i.e. the maintenance of a viable population as well as a population collapse) are possible (figure 3*b*). The outcome

depends on the rate of the increase in environmental stress $\varepsilon$ and on the magnitude of $E_\text{max}$. Specifically, the faster the increase and the closer $E_\text{max}$ to the tipping point when evolution occurs (TPc$_\text{ecoevo}$), the more likely a regime shift is. When the regime shift occurs, it causes a permanent collapse because the system moves from one stable state to another.

The dynamics of the evolving population in the transition region contrasts with the dynamics of a non-evolving population, because in the latter a collapse always occurs when $E_\text{max}$ exceeds the tipping point that marks the collapse in the absence of evolution (TPc$_\text{eco}$). Therefore, a non-evolving population is not sensitive to the rate of change of environmental stress.

In the transition region, the occurrence of the regime shift causing the collapse of an evolving population also depends on genetic trait variance (figure 4). High genetic trait variance prevents the collapse of the population even when environmental stress increases fast (figure 4*a*). This is because high genetic trait variance enables a fast increase in the mean trait (right top corner in figure 4*b*), and the critical threshold of environmental stress, where the tipping point of collapse occurs, depends on the population mean trait, $\bar{x}$, (figure 4*c*) as revealed by the analytical investigation of the system:

$$E_\text{TPc} = \bar{x} + \sqrt{2(\sigma^2 + \tau^2)} \sqrt{-\ln\left(\frac{4K\sqrt{\sigma^2 + \tau^2}\,(\mu_0 + \mu_1(\sigma^2 + \bar{x}^2))}{b_\text{max}\,\tau(A - K)^2}\right)}.$$

(2.1)

This expression shows that when the mean trait increases, the environmental stress level at which the tipping point of collapse occurs also increases (a detailed analytical investigation of the effect of mean trait on the location of this tipping point can be found in electronic supplementary material, appendix C). Therefore, fast adaptation enabled by high genetic trait variance causes the shift of the tipping point to occur fast, preventing a regime shift and thus a population collapse. This preventive effect due to fast evolutionary change is reduced by increasing levels of the maximum environmental stress $E_\text{max}$ experienced by the population (electronic supplementary material, figure S1).

In contrast to the configuration of the equilibrium states observed in figure 2, the two bistability regions may not overlap (blue region when evolution does not take place and red when it occurs in figure 5*a*). In this case, the occurrence of a regime shift causing the collapse of an evolving population depends on the rate of the increase in environmental stress and on the magnitude of $E_\text{max}$ in the same manner than when the bistability regions overlap (figure 5*b* versus figure 3*a*). However, the dynamics after the collapse differ because in the transition region (TPc$_\text{eco} < E_\text{max} <$ TPc$_\text{ecoevo}$) two qualitatively distinct regions occur:

— One region corresponds to an evolving population experiencing gradual increase in environmental stress to a maximum level $E_\text{max}$ above the tipping point that marks the invasion threshold (TPi$_\text{ecoevo}$) and the tipping point that marks the collapse (TPc$_\text{ecoevo}$) when evolution is enabled by genetic trait variance. In this region ($2.46 < E_\text{max} < 5.19$ in figure 5*a,b*), the two eco-evolutionarily stable states exist: persistence and extinction. Therefore,

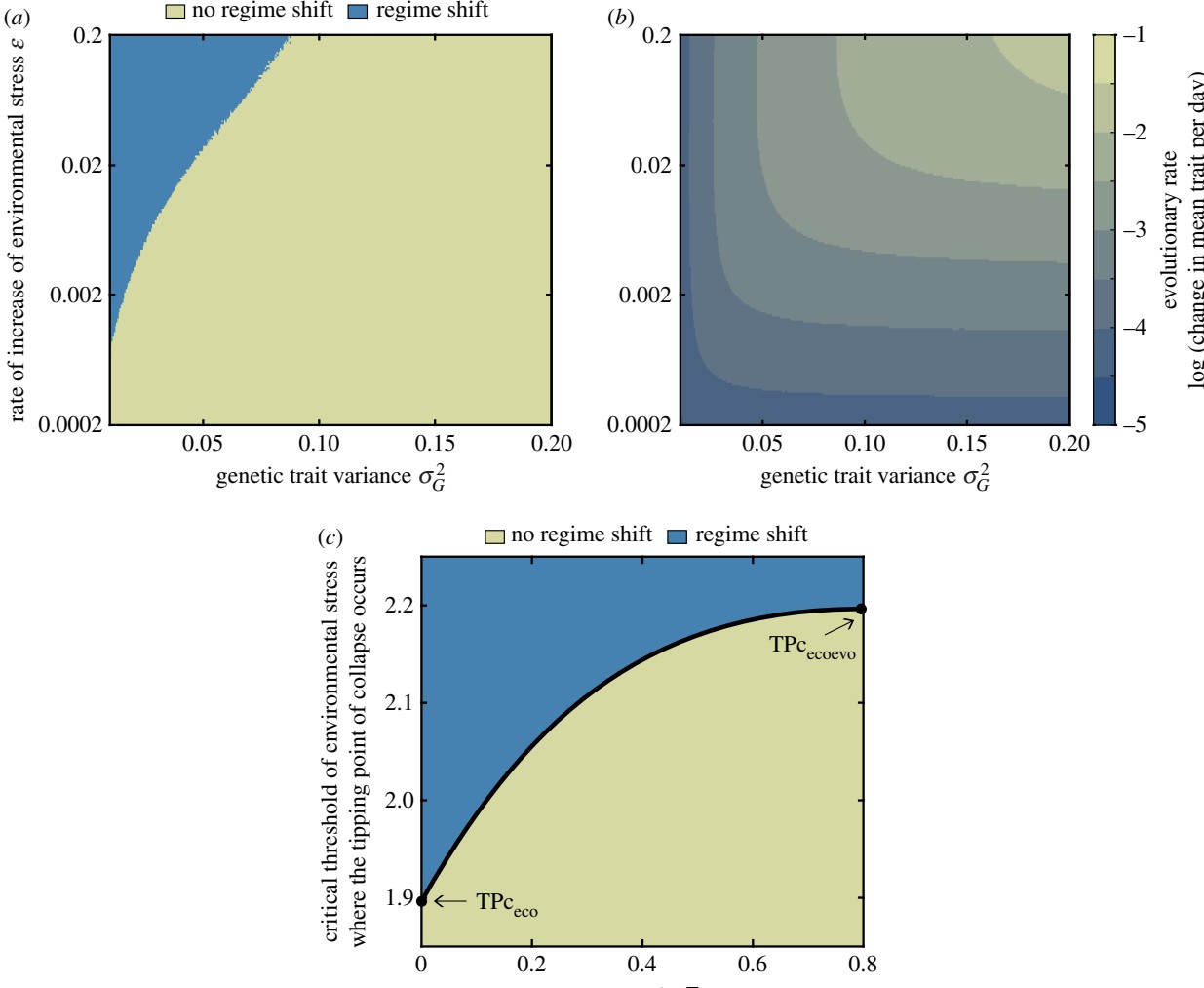

**Figure 4.** (a) Occurrence of a regime shift in an evolving population subjected to Allee effect as a function of the genetic trait variance and the rate of the increase in environmental stress. (b) Evolutionary rate of the populations simulated in (a) during increasing environmental stress (measured as the difference between the change in mean trait of the population during the time it experiences increasing stress and divided by the length of this time period). (c) Critical threshold of environmental stress where the tipping point that marks the collapse (i.e. regime shift) occurs as a function of the mean trait value of the population (see equation (2.1), the analytical derivation of this function is in electronic supplementary material, appendix C). Parameter values as in figure 2. Additional parameters in (a) and (b): $E_{max} = 2$; and in (c): $\sigma_G^2 = \sigma^2 = 0.05$. (Online version in colour.)

once a regime shift occurs, the population collapse is permanent because the system moves from one stable state to another. This is the same behaviour found in the transition region when the two bistability regions overlap (e.g. $1.95 < E_{max} < 2.2$ in figure 3a).

— Another region corresponds to an evolving population experiencing gradual increase in environmental stress to a maximum level $E_{max}$ above the tipping point that marks the collapse in the absence of evolution ($TPc_{eco}$) and below the tipping point that marks the invasion threshold when evolution is enabled by genetic trait variance ($TPi_{ecoevo}$). In this region ($1.95 < E_{max} < 2.46$ in figure 5a,b), only one eco-evolutionarily stable state exists, that is the persistence state. Interestingly, fast increase in environmental stress results in a regime shift causing a collapse even though the only eco-evolutionarily stable state is that in which the population persists (striped region in figure 5b). However, this collapse is transient because extinction is not an eco-evolutionarily stable state. The population thus can recover after the transient collapse without need of intervention to reduce environmental stress (S2 in figure 5c).

In the absence of evolution, the extinct state is stable in the transition region. Therefore, a transient collapse cannot occur, and the recovery of a non-evolving population requires a reduction in environmental stress to a level below the tipping point that marks the invasion threshold in the absence of evolution ($TPi_{eco}$).

Although the extinct state is eco-evolutionarily unstable (eco-evolutionary state in the region $1.95 < E_{max} < 2.46$ in figure 5a), the capacity of the population to invade (i.e. to have positive growth rate at low density) depends not only on the level of environmental stress, but also on the mean trait value (figure 6). Therefore, if evolutionary trait changes are not possible due to the lack of genetic trait variance, a population cannot grow from low density when the combination of its trait value and the level of environmental stress results in a negative *per capita* growth rate (figure 6a). For instance, when environmental stress is 2, a population cannot invade if its trait value is smaller than 0.98 (e.g. a population with trait value 0 or 0.5 does not recover in figure 6; in both cases the *per capita* growth rate is negative in figure 6a); but it can invade if its trait value is larger (e.g.

royalsocietypublishing.org/journal/rspb　Proc. R. Soc. B **288**: 20211192

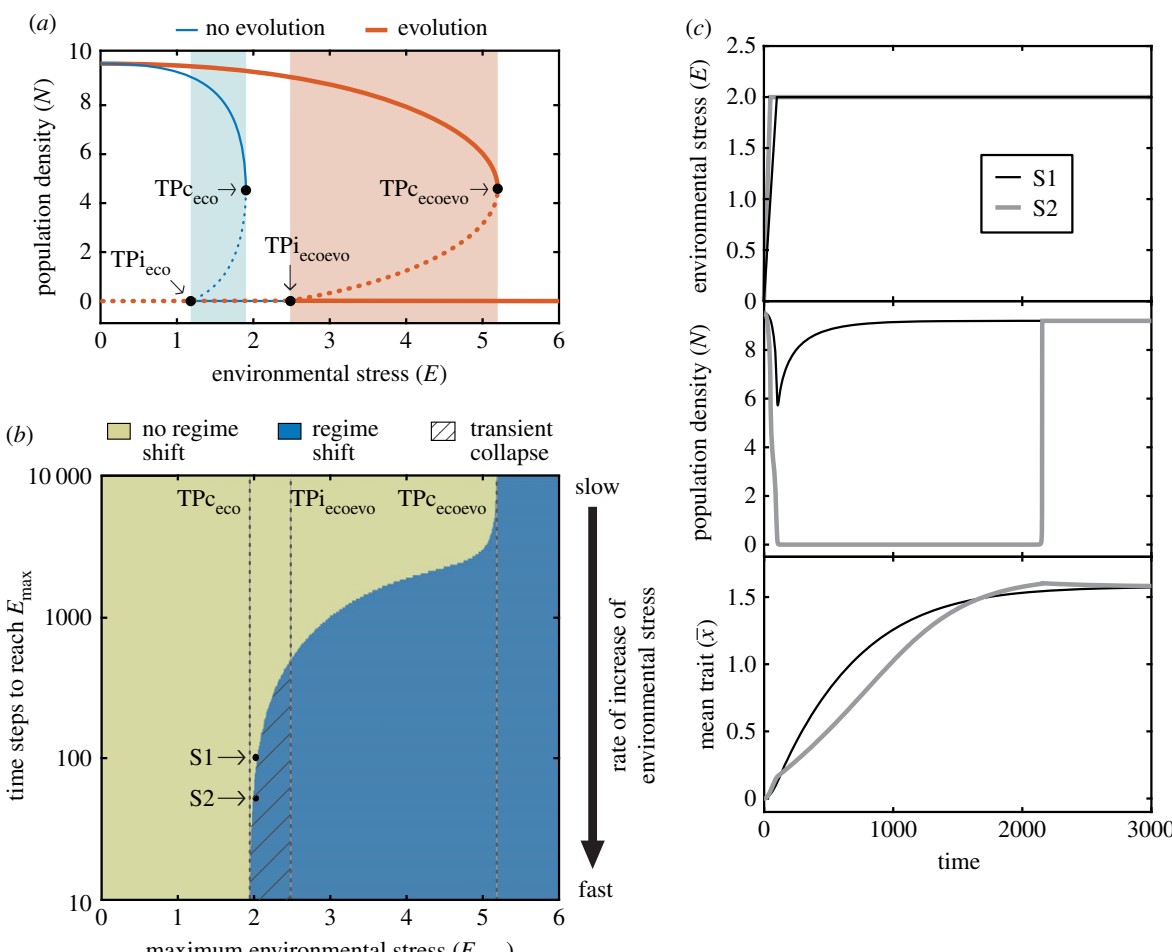

**Figure 5.** (a) Population density in the equilibrium as a function of fixed levels of environmental stress when evolution does not take place (due to the absence of genetic trait variance, $\sigma_G^2 = \sigma^2 = 0$; blue lines) and when it does occur ($\sigma_G^2 = \sigma^2 = 0.05$; red lines). Solid lines represent stable equilibrium states and dotted lines represent unstable equilibrium states. Bistability occurs in the shaded region (blue when there is no evolution, and red when evolution takes place). Tipping points mark the transition between ASSs (black dots; $TPc_{eco}$, $TPc_{ecoevo}$ correspond to the collapse threshold and $TPi_{eco}$, $TPi_{ecoevo}$ to the invasion threshold). (b) Occurrence of a regime shift in an evolving population subjected to Allee effect as a function of the maximum environmental stress and the rate of the increase in environmental stress (i.e. time steps—days—required to reach the maximum environmental stress level). Vertical dashed lines indicate the values of the tipping points in panel (a) ($TPc_{eco}$, $TPc_{ecoevo}$, $TPi_{ecoevo}$). (c) Environmental stress (top panel), population density (middle panel) and mean trait value (bottom panel) dynamics in the simulations indicated by the points S1 and S2 in panel (b). Parameter values in (a–c): $K = 10$, $A = -1$, $b_{max} = 1$, $\mu_0 = 0.5$, $\mu_1 = 0.1$, $\tau = 1$. Additional parameters in (b): $\sigma_G^2 = \sigma^2 = 0.05$; and in (c): $E_{max} = 2$, $\sigma_G^2 = \sigma^2 = 0.05$, $\varepsilon = 0.02$ in S1, and $\varepsilon = 0.04$ in S2 (S1 and S2 only differ in the rate of the increase in environmental stress $\varepsilon$). (Online version in colour.)

a population with trait value 1, 1.5 or 2 recovers in figure 6b; in all cases the *per capita* growth rate is positive in figure 6a).

As shown in figure 5c, the recovery after a transient collapse is theoretically possible because the fitness gradient and the genetic trait variance are positive, and in the model framework used here (quantitative genetics), this is enough to enable evolution towards a larger trait value. However, evolution cannot occur in an extinct population in the wild. Hence, if a population becomes extinct following a regime shift, reinvasion can only occur due to immigration. Nonetheless, the trait-dependent capacity of invasion described above implies that population recovery is not possible if invading individuals have the same trait value of that of the extinct population. Instead, the population can recover and reach its eco-evolutionarily stable state only if immigration of individuals with larger trait values occurs.

## (d) Community level: predator–prey interaction with stage-structured prey

Predator species shape the structure of their prey populations, but disturbances in predator abundance can render the predator population incapable of shaping the prey size structure [22]. As a consequence, changes in predator mortality or productivity may result in a regime shift from a state of high density to an alternative state of low density of the predator population [22]. This phenomenon has caused dramatic trophic cascades in aquatic ecosystems [23] and collapses of fisheries with devastating socioeconomic impacts [24]. I use the simplest description of the system [25] and incorporate predator evolution to explore the eco-evolutionary consequences of environmental stress at the community level. A detailed model description can be found in the electronic supplementary material, appendix B.

I analyse the predator–prey system following the same procedure used in the analysis of the demographic Allee effect: I first perform a stability analysis of the system at fixed levels of environmental stress, and subsequently, analyse the system dynamics of an evolving predator population that experiences increasing levels of environmental stress. Because the results are qualitatively the same as those of the population subjected to a demographic Allee effect, I do not present them here in detail but discuss them in the following section 'General

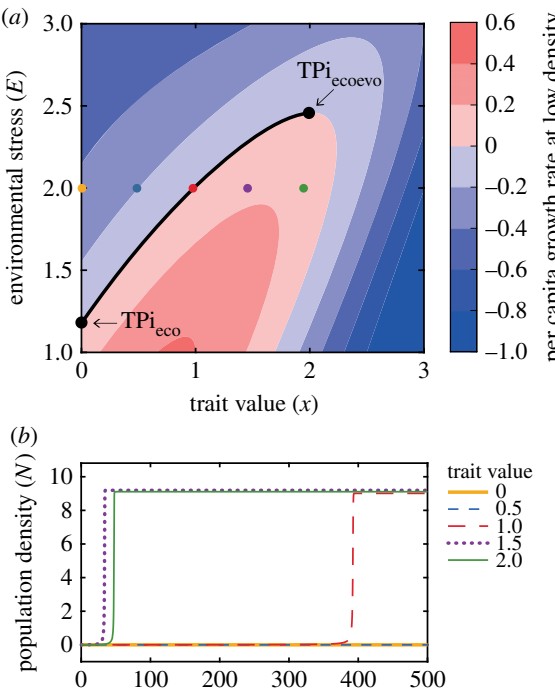

**Figure 6.** (*a*) *Per capita* growth rate at low density ($N = 10^{-6}$) as a function of fixed levels of environmental stress and trait value (calculated using equation A1.3, see electronic supplementary material, appendix A). The solid black line indicates the critical threshold of environmental stress where the tipping point of invasion occurs (the minimum corresponds to TPi$_{eco}$ and the maximum to TPi$_{ecoevo}$ as shown in figure 5(*a*); the analytical derivation of this function is in electronic supplementary material, appendix C). (*b*) Density dynamics of a population whose individuals are identical ($\sigma_G^2 = \sigma^2 = 0$; no trait variance, and thus no evolution) and their phenotype is indicated by the legend. The population is seeded with an initial density of 0.001 and experiences a constant environmental stress level $E = 2$. The *per capita* growth rate of each population is indicated in panel (*a*) by the dots of the same colour. Parameter values as in figure 5*a*. (Online version in colour.)

patterns in eco-evolutionary systems with ASSs' (a detailed results description can be found in electronic supplementary material, appendix D).

## (e) Ecosystem level: shallow lake

The most studied example of regime shifts in natural ecosystems is probably the regime shift between a clear-water and a turbid state in shallow lakes [26]. Nutrient loading can shift the ecosystem from the clear-water to the turbid state dominated by algae (i.e. phytoplankton). The competitive interaction between macrophytes and algae is key in the existence of ASSs in shallow lakes. I use the simplest ecological model of the shallow lake ecosystem [26] and incorporate macrophyte evolution to explore the eco-evolutionary consequences of environmental stress (i.e. nutrient loading) at the ecosystem level. A detailed model description can be found in the electronic supplementary material, appendix B.

I analyse the shallow lake system following the same procedure used in the analysis of the demographic Allee effect: I first perform a stability analysis of the system at fixed levels of environmental stress (i.e. nutrient loading), and subsequently, analyse the system dynamics of an evolving

macrophyte population that experiences increasing levels of environmental stress. Because the results are qualitatively the same as those of the population subjected to a demographic Allee effect and the predator–prey interaction, I do not present them here in detail but discuss them in the following section 'General patterns in eco-evolutionary systems with ASSs' (a detailed results description can be found in electronic supplementary material, appendix D).

## (f) General patterns in eco-evolutionary systems with ASSs under stress

I investigate how ecological, evolutionary and environmental stress dynamics influence the occurrence of regime shifts in three different ecological systems with ASSs corresponding to different ecological scales: population, community and ecosystem levels. Despite assumptions underlying bistability and phenotypic responses to environmental stress greatly differing among the systems, I identified general patterns across them in their stability (asymptotic behaviour) as well as their transient dynamics.

The stability (i.e. asymptotic behaviour) analyses reveal that the presence of genetic trait variance and thus of evolution shifts the tipping points that mark a regime shift to a different level of environmental stress than that expected in the absence of evolution. In the ecological systems investigated here, evolutionary changes shift the tipping points to higher environmental stress level, increasing the resilience of the systems. However, the transient behaviour of the systems in response to a deteriorating environment cannot be predicted based only on their asymptotic behaviour. Indeed, in the three systems, fast environmental change causes a regime shift before environmental stress exceeds the tipping point predicted by the eco-evolutionary stability analysis. Remarkably, this regime shift can even occur at levels of environmental stress in which the system has only one globally stable state, and still, the system may temporarily collapse into an 'unstable' state. When this occurs, the system may recover to the original (and only stable) state, however, its recovery depends on the phenotype of the organisms in the system. Interestingly, such regime shifts may be prevented by high genetic trait variance enabling a fast evolutionary process.

## 3. Discussion

This study shows that eco-evolutionary feedbacks can qualitatively affect the response of ecological systems with ASSs to increasing environmental stress. In the absence of these feedbacks, the ecological systems investigated are sensitive only to the magnitude of environmental stress, hence, a regime shift can occur only when this magnitude exceeds a tipping point. The introduction of eco-evolutionary feedbacks makes these ecological systems also sensitive to the rate at which environmental changes occur. As a consequence, an increase in environmental stress to a certain magnitude can have two different outcomes: the system remains in its current state, or it undergoes a regime shift. Whether a regime shift occurs depends on both the rate at which environmental stress increases and the rate of evolution. Specifically, when environmental stress increases slowly and genetic trait variance enables a fast evolutionary process, environmental change does not cause a regime shift. Conversely, when

environmental stress increases quickly and the evolutionary process is slow, a regime shift occurs.

Most ecological theory devoted to the study of critical transitions has considered regime shifts that occur when the magnitude of an environmental stressor exceeds a tipping point (e.g. [27–29]). The focus of ecological theory on this mechanism, known as bifurcation-tipping, has led ecologists to set management goals based on maintaining the magnitude of environmental stressors below the tipping point to prevent regime shifts [30]. However, I find that a regime shift can occur at environmental stress levels below the tipping point when environmental change is fast. This finding joins recent theoretical studies that have also shown that ecological systems may be sensitive not only to the magnitude of environmental changes but also to the rate at which changes are imposed, and thus that establishing a target magnitude of an environmental stressor might be insufficient to prevent regime shifts in ecosystems [31,32]. Different from these studies, here I identified the eco-evolutionary feedback as a mechanism that can underlie such sensitivity.

In the three ecological systems investigated here, the sensitivity to the rate, and not just the magnitude, of environmental change is caused by the different time scales at which ecological and evolutionary dynamics occur. This kind of regime shifts, also known as rate-induced critical transitions (or rate-tipping), was hypothesized to be a general property of dynamical systems that have processes operating on different time scales (fast–slow systems) [33]. In line with this theory, I find a higher occurrence of regime shifts with increasing separation of the time scales at which ecological and evolutionary processes occur. In the models investigated here, genetic trait variance determines the time scale separation between the ecological and evolutionary dynamics. When genetic trait variance is high, natural selection can induce the trait changes required to shift the tipping point to a higher stress level quickly relative to the ecological processes, preventing the system to cross the threshold and thus a regime shift. Conversely, when genetic trait variance is low, the evolutionary process that shifts the tipping point to a higher stress level occurs slowly relative to the ecological processes, increasing the occurrence of regime shifts. Besides standing variation, other properties of the ecosystem and their interacting species can influence the speed of evolution, probably producing the same dynamics described here. For instance, slower evolutionary dynamics are expected in species with longer generation times; hence, I would expect a higher occurrence of regime shifts in these species. Interestingly, long-term abundance time series of 55 different taxa show that critical transitions are more likely to occur in species with longer generation time [34].

Ecological theory has considered regime shifts in which ecosystems move from one stable state to another. As a consequence, after a regime shift, the ecosystem persists in the alternative stable state if environmental stress remains at this level [28]. Here I find that regime shifts, specifically rate-induced critical transitions, may occur at levels of environmental stress in which the ecosystem has only one globally stable state, and yet it can shift to an 'unstable' state. Remarkably, a switch back to the initial and only stable ecosystem state can occur despite the level of environmental stress remains unchanged so long as the evolutionary process enables phenotypic changes. This is because divergence in the time scales of coupled ecological and evolutionary responses can

produce dynamic behaviors on short time scales that cannot be anticipated by studying only the asymptotic stability of the system [35]. Therefore, the stability (bifurcation) analysis, commonly applied to determine the presence of ASSs that set the base for management targets, is insufficient to predict the occurrence of these regime shifts.

A large body of ecological theory has investigated how evolutionary changes induced by natural selection can rescue populations experiencing environmental stress from extinction [36,37]. Here, I identify a mechanism whereby such rescue may occur. Specifically, in the study case of a demographic Allee effect, I show that the extinction state, which in the absence of evolution is an attractor for the demographic dynamics, can become unstable when evolution takes place. Therefore, populations experiencing stress may recover following a decline as a consequence of the instability that the evolutionary process introduces to the system dynamics. This finding may be relevant not only for the management of endangered populations but also of non-native populations as it suggests that their establishment may, in some cases, depend on their capacity to evolve, or evolvability. In fact, evolutionary change has been shown to increase the potential for growth of small introduced populations [38].

Perhaps one of the biggest challenges will be to empirically study how eco-evolutionary feedbacks make ecosystems sensitive to the rate of environmental change. A first attempt may use an approach combining models and data. For instance, a recent study, using a model whose parameters were estimated from data, showed that behavioural feedbacks may cause rate-induced regime shifts in coral reefs [31]. However, determining whether a regime shift observed in the past was caused by the rate of change of an environmental condition may be challenging due to the lack of data on the effects of the rate of change in ecosystems [39]. Furthermore, documented regime shifts at the ecosystem scale rarely include information about trait changes of the organisms involved [10], which is necessary to study the feedbacks between ecological and evolutionary processes. Empirical support for the occurrence of shifts in ecosystem state induced by the rate of change of an environmental condition arising from manipulative experimental studies may be more conclusive. Several experimental approaches have been designed to study the feedbacks between ecological and evolutionary processes [13,40]. These could be combined with approaches used to experimentally investigate alternative ecological states [41,42] to test the occurrence of rate-induced regime shifts.

In this study, I consider the evolution of the species that is abundant in the ecological state present at low levels of environmental stress. In this case, the tipping point that marks the transition to the ASS is shifted to a higher level of environmental stress. Perhaps, the evolution of antagonistic species (e.g. defense traits in the prey in the predator–prey system, or competitive traits in algae in the shallow lake ecosystem) may shift the tipping points to a lower stress level. As in the cases studied here, I expect rate-induced regime shifts to occur in evolving ecological systems when environmental stress increases to levels in between the tipping point set by the absence of evolution and the tipping point set by the presence of evolution (transition region between $TPc_{eco}$ and $TPc_{ecoevo}$ in the results). Therefore, if evolution in antagonistic species shifts the tipping point to a lower environmental stress level, rate-induced regime shifts may occur before stress levels exceed the tipping

point of the ecological system without evolution. Predicting the effects of co-occurring evolutionary processes of multiple interacting species on ecosystem tipping points and their associated dynamics will be more challenging, especially when evolutionary rates greatly differ due to, for instance, generation time differences among the species. Further research is therefore needed to gain insights into the responses to environmental change of ecosystems with coevolving species to inform management strategies.

Simultaneous ecological and evolutionary responses to environmental changes are common in ecosystems [2]. It will therefore prove necessary to gain further insight into the interaction between ecological, evolutionary and environmental stress dynamics to understand how ecosystems respond to environmental change. The present study reveals that the interaction between ecological and evolutionary dynamics makes ecosystems sensitive not only to the magnitude of environmental changes but also to the rate at which changes are imposed. These findings are general across the studied systems despite their differences in ecological scales and underlying

assumptions, and thus extend well beyond these systems. Although rate-induced regime shifts have not been documented in natural ecosystems, they might be very common because concurrent ecological and evolutionary changes in response to environmental changes are widespread in nature [2]. Here, I have shown that rates of environmental change and evolution play a crucial role in the occurrence of rate-induced regime shifts. These results suggest that preventing these catastrophic transitions is more likely in a scenario of slow environmental change and fast phenotypic change. This highlights the need to mitigate the effects of environmental change and to conserve phenotypic diversity in ecosystems.

Data accessibility. This article has no additional data.
Competing interests. I declare I have no competing interests.
Funding. This research was supported by the Swiss National Science Foundation (SNSF) under the Spark funding scheme no. 190940.
Acknowledgements. The author gratefully acknowledges Gregory Roth, Nele Schuwirth, Carlo Albert and Carlos Melian for helpful feedback on this work.

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
