## [Peer Review File · Proceedings of the Royal Society B: Biological Sciences]

Review History

RSPB-2021-0277.R0 (Original submission)

Review form: Reviewer 1

Recommendation

Major revision is needed (please make suggestions in comments)

Scientific importance: Is the manuscript an original and important contribution to its field?

Excellent

General interest: Is the paper of sufficient general interest?

Good

Quality of the paper: Is the overall quality of the paper suitable?

Good

Is the length of the paper justified?

Yes

Should the paper be seen by a specialist statistical reviewer?

No

Do you have any concerns about statistical analyses in this paper? If so, please specify them explicitly in your report.

No

It is a condition of publication that authors make their supporting data, code and materials available - either as supplementary material or hosted in an external repository. Please rate, if applicable, the supporting data on the following criteria.

Is it accessible?

No

Is it clear?

No

Is it adequate?

No

Do you have any ethical concerns with this paper?

No

Comments to the Author

In this paper, the authors construct general theoretical models to examine potential interactions among ecological dynamics, rates of phenotypic evolution, and rates of environmental change across three contexts: a population with an Allee effect, a predator-prey system, and an shallow lake ecosystem. Across these contexts, they find that rates of environmental change can, themselves, determine the state of the system, with faster rates of change capable of causing state shifts that are precluded at slower paces of change. They further find that the pace of evolution of organismal phenotypes that dictate adaptability to environmental conditions can counteract this effect, with faster paces of evolution allowing systems to avoid state shifts by essentially keeping pace with environmental change. This work provides new evidence for the importance of both phenotypic diversity and eco-evolutionary feedbacks to both our knowledge and sustainable management of natural ecological systems.

I found this to be an interesting and well written paper with novel insights that will appeal to a broad range of evolutionary biologists and ecologists. This work contributes important evolutionary extensions to recent findings of rate dependence in ecological regime shifts. My main issue with the paper, as written, is that it fails to appropriately acknowledge recent discoveries and, thus, to place the current findings in the context of the literature. Two of the major findings discussed – that evolutionary feedbacks can qualitatively determine whether regime shifts will occur in ecological systems subject to different rates of environmental change, and that, in such cases, regime shifts can occur even when there is one globally stable equilibrium – are novel, and, alone, make this an important contribution. The rest of the major findings – that the state of ecological systems can be sensitive, qualitatively, to the pace and not just magnitude of environmental change and that standard equilibrium analyses can, therefore, mislead – were recently reported in a paper that the authors cite, but without reference to this fact.

For example, the opening paragraph the Discussion, summarizing the major contribution of this work, fails to acknowledge, and actually contradicts the fact that there is existing ecological theory - cited later in the Discussion - (Gil et al. PNAS 2020) that does posit that stability analysis is insufficient to predict critical thresholds at which regime shifts occur when feedback loops influence transient dynamics. This previous study looked specifically at the interaction between ecological dynamics and rates of environmental change. They did not look at the additional component of differing rates of phenotypic evolution, which is probed in the current manuscript. This is the point of novelty of this manuscript, which corroborates and builds off of the major

findings of Gil et al. The Discussion should be revised accordingly, including careful qualification of what specifically sets the current ms apart from this other highly related recent work.

Furthermore, the conclusions discussed on lines 353-364 are precisely the same findings revealed by in Gil et al. 2020 (with the exception of the sentence on lines 356-358, which is, indeed, a wholly novel finding). Accordingly, this paper should be clearly acknowledged to clarify that these findings align with those of past work (rather than implying that these are entirely novel findings, as the present presentation does). I suggest carefully reading that paper and revising accordingly.

The paragraph that follows (beginning on Lines 365-366) would also benefit from better clarifying the state of the field: Gil et al. 2020 showed that rate-induced regime shifts are possible not in a general model, non system specific model (as in this ms) but in a model parameterized based on data from a real ecosystem: a tropical coral reef. Thus, importantly, Gil et al. has gone a step further than general theory and exemplifies the potential for the kind of rate-dependence emerging from separations in timescales (shown in the current ms) to be an important feature of natural ecosystems.

More specific issues/revisions (fig or line number noted, where appropriate):

Fig. 2: In lower panel, I was struck by the nonlinearity in trait change over env stress - why the acceleration? Is this qualitative shape sensitive to the value of τ ?

Fig. 3: Is panel A from a model with evolution? I assume so, since that threshold lines up with the bifurcation. If so, this suggests Fig. 2 corresponds to a case when env stress was ramped up from zero quite slowly (over $>10,000$ days), but it's unclear if/how env stress ramping was implemented in Fig. 2. If there was no ramping and, instead, a fixed value was used for env stress, wouldn't that correspond to a much lower threshold (i.e., an immediate increase to the max level would be the fastest possible)? Please clarify.

In addition, I would clarify in the beginning of caption that this is an "evolving population". Also, it's a bit odd that the trait keeps evolving after the population has collapsed. This makes me wonder: presumably, a small amount of the population persisting, if the trait continues to evolve after the max env change is reached, ought to recover, right? Did you check to see if this was possible? In other words, what prevents the population from bouncing back, as trait values continually improve over time in fig 3, like fig 5? Please clarify in the main text.

Fig 4C:

I was struck that this effect was only $\sim 18\%$ between thresholds of vastly different trait values. It would be helpful to know what range of effect sizes you observe across parameter values and ecological contexts explored.

Fig. 5: I'd recommend aligning the x axes of panels a and b. Also, I would clarify at the start of the caption that this is under the assumption of different rates of environmental change, and I would clarify which rate of change (e.g., in time steps to reach E_{max}) these solutions pertain to.

5A: are we looking at different rates of env. stressor increase between the evo and ecoevo solutions? Panel B suggests yes, since the TP1eco threshold seems to correspond to a very fast rate of stressor change, while TP1ecoeco seems to pertain to a very slow rate of stressor change. Please clarify.

Fig. 6:

I found Figure 6B challenging to unpack, since it combines many elements in a single visual. Part of what makes this confusing is that thresholds for both the eco and ecoevo models are shown on a 3D plot for which trait values appear fixed (non-evolving). I recommend providing further details in the main text that step the reader through the panel (or breaking it down into simpler sub components). Additional points of clarification should also address the following:

- It appears that the numerical solutions for the ecoevo feedbacks model presented in Fig. 6A

pertain to a trait value of ~ 1.9 , according to panel B. Does this mean, under the specified parameterization, that the ecoevo model evolved to achieve this level of the trait (which is being shown as a fixed trait value in panel B)? I would clarify and, in any case, I think it would be helpful to point out this link to help highlight how the different panels correspond to one another.

- Can a transient population collapse occur for the model with no evolution? Since the trait value that contributed to the collapse cannot change in this case, it seems the answer is 'no'. Yet, Fig. 6B notes that the vertical lines (the left of which pertains to TP2eco) delimit regions where transient collapse can occur. Please clarify.

- Why don't the TP2eco values correspond between panels a and b, as they do for TP2ecoevo?

- I'm wondering: would a line that bisects the unimodal curves in the per capita growth rate isoclines delimit the optimal fixed trait values across stress levels? If so, I would include this too.

122: 2nd sentence: "and" should be "or"

Fig 2: change "being" to "go"

153: I would help the reader by setting this section off with another subheading that clarifies that all of the results described therein pertain to the exploration of the effect of varying the rate at which the environmental stress changes (and, correspondingly, I'd add a sub heading for the preceding section that clarifies that this section deals with evolutionary response to stress (which is fixed in magnitude)

276: algae should be alga

304: differ should be differing

Review form: Reviewer 2

Recommendation

Major revision is needed (please make suggestions in comments)

Scientific importance: Is the manuscript an original and important contribution to its field?

Excellent

General interest: Is the paper of sufficient general interest?

Good

Quality of the paper: Is the overall quality of the paper suitable?

Marginal

Is the length of the paper justified?

Yes

Should the paper be seen by a specialist statistical reviewer?

No

Do you have any concerns about statistical analyses in this paper? If so, please specify them explicitly in your report.

No

It is a condition of publication that authors make their supporting data, code and materials available - either as supplementary material or hosted in an external repository. Please rate, if applicable, the supporting data on the following criteria.

Is it accessible?

No

Is it clear?

No

Is it adequate?

No

Do you have any ethical concerns with this paper?

No

Comments to the Author

See attached file (See Appendix A)

Decision letter (RSPB-2021-0277.R0)

04-May-2021

Dear Ms Chaparro Pedraza:

I am writing to inform you that your manuscript RSPB-2021-0277 entitled "Fast environmental change and eco-evolutionary feedbacks can drive regime shifts in ecosystems before tipping points are crossed" has, in its current form, been rejected for publication in Proceedings B.

This action has been taken on the advice of referees, who have recommended that substantial revisions are necessary. With this in mind we would be happy to consider a resubmission, provided the comments of the referees are fully addressed. However please note that this is not a provisional acceptance.

Sincerely,
Dr Locke Rowe
mailto: proceedingsb@royalsociety.org

Associate Editor
Board Member: 1
Comments to Author:

Dear Dr Pedraza

as you will see from the detailed feedback provided, both reviewers are very positive about your work and agree that it would ultimately merit publication. However, both stress that more needs to be done to place your study and results in the context of the existing literature, including the recently published work by Gil et al. in PNAS (2020).

Reviewer(s)' Comments to Author:

Referee: 1

Comments to the Author(s)

In this paper, the authors construct general theoretical models to examine potential interactions among ecological dynamics, rates of phenotypic evolution, and rates of environmental change across three contexts: a population with an Allee effect, a predator-prey system, and an shallow lake ecosystem. Across these contexts, they find that rates of environmental change can, themselves, determine the state of the system, with faster rates of change capable of causing state shifts that are precluded at slower paces of change. They further find that the pace of evolution of organismal phenotypes that dictate adaptability to environmental conditions can counteract this effect, with faster paces of evolution allowing systems to avoid state shifts by essentially keeping pace with environmental change. This work provides new evidence for the importance of both phenotypic diversity and eco-evolutionary feedbacks to both our knowledge and sustainable management of natural ecological systems.

I found this to be an interesting and well written paper with novel insights that will appeal to a broad range of evolutionary biologists and ecologists. This work contributes important evolutionary extensions to recent findings of rate dependence in ecological regime shifts. My main issue with the paper, as written, is that it fails to appropriately acknowledge recent discoveries and, thus, to place the current findings in the context of the literature. Two of the major findings discussed – that evolutionary feedbacks can qualitatively determine whether regime shifts will occur in ecological systems subject to different rates of environmental change, and that, in such cases, regime shifts can occur even when there is one globally stable equilibrium – are novel, and, alone, make this an important contribution. The rest of the major findings – that the state of ecological systems can be sensitive, qualitatively, to the pace and not just magnitude of environmental change and that standard equilibrium analyses can, therefore, mislead – were recently reported in a paper that the authors cite, but without reference to this fact.

For example, the opening paragraph the Discussion, summarizing the major contribution of this work, fails to acknowledge, and actually contradicts the fact that there is existing ecological theory - cited later in the Discussion - (Gil et al. PNAS 2020) that does posit that stability analysis is insufficient to predict critical thresholds at which regime shifts occur when feedback loops influence transient dynamics. This previous study looked specifically at the interaction between ecological dynamics and rates of environmental change. They did not look at the additional component of differing rates of phenotypic evolution, which is probed in the current manuscript. This is the point of novelty of this manuscript, which corroborates and builds off of the major

findings of Gil et al. The Discussion should be revised accordingly, including careful qualification of what specifically sets the current ms apart from this other highly related recent work.

Furthermore, the conclusions discussed on lines 353-364 are precisely the same findings revealed by in Gil et al. 2020 (with the exception of the sentence on lines 356-358, which is, indeed, a wholly novel finding). Accordingly, this paper should be clearly acknowledged to clarify that these findings align with those of past work (rather than implying that these are entirely novel findings, as the present presentation does). I suggest carefully reading that paper and revising accordingly.

The paragraph that follows (beginning on Lines 365-366) would also benefit from better clarifying the state of the field: Gil et al. 2020 showed that rate-induced regime shifts are possible not in a general model, non system specific model (as in this ms) but in a model parameterized based on data from a real ecosystem: a tropical coral reef. Thus, importantly, Gil et al. has gone a step further than general theory and exemplifies the potential for the kind of rate-dependence emerging from separations in timescales (shown in the current ms) to be an important feature of natural ecosystems.

More specific issues/revisions (fig or line number noted, where appropriate):

Fig. 2: In lower panel, I was struck by the nonlinearity in trait change over env stress - why the acceleration? Is this qualitative shape sensitive to the value of τ ?

Fig. 3: Is panel A from a model with evolution? I assume so, since that threshold lines up with the bifurcation. If so, this suggests Fig. 2 corresponds to a case when env stress was ramped up from zero quite slowly (over $>10,000$ days), but it's unclear if/how env stress ramping was implemented in Fig. 2. If there was no ramping and, instead, a fixed value was used for env stress, wouldn't that correspond to a much lower threshold (i.e., an immediate increase to the max level would be the fastest possible)? Please clarify.

In addition, I would clarify in the beginning of caption that this is an "evolving population".

Also, it's a bit odd that the trait keeps evolving after the population has collapsed. This makes me wonder: presumably, a small amount of the population persisting, if the trait continues to evolve after the max env change is reached, ought to recover, right? Did you check to see if this was possible? In other words, what prevents the population from bouncing back, as trait values continually improve over time in fig 3, like fig 5? Please clarify in the main text.

Fig 4C:

I was struck that this effect was only ~18% between thresholds of vastly different trait values. It would be helpful to know what range of effect sizes you observe across parameter values and ecological contexts explored.

Fig. 5: I'd recommend aligning the x axes of panels a and b. Also, I would clarify at the start of the caption that this is under the assumption of different rates of environmental change, and I would clarify which rate of change (e.g., in time steps to reach E_{max}) these solutions pertain to.

5A: are we looking at different rates of env. stressor increase between the evo and ecoevo solutions? Panel B suggests yes, since the TP1eco threshold seems to correspond to a very fast rate of stressor change, while TP1ecoeco seems to pertain to a very slow rate of stressor change. Please clarify.

Fig. 6:

I found Figure 6B challenging to unpack, since it combines many elements in a single visual. Part of what makes this confusing is that thresholds for both the eco and ecoevo models are shown on a 3D plot for which trait values appear fixed (non-evolving). I recommend providing further details in the main text that step the reader through the panel (or breaking it down into simpler sub components). Additional points of clarification should also address the following:

- It appears that the numerical solutions for the ecoevo feedbacks model presented in Fig. 6A pertain to a trait value of ~ 1.9 , according to panel B. Does this mean, under the specified parameterization, that the ecoevo model evolved to achieve this level of the trait (which is being shown as a fixed trait value in panel B)? I would clarify and, in any case, I think it would be helpful to point out this link to help highlight how the different panels correspond to one another.

- Can a transient population collapse occur for the model with no evolution? Since the trait value that contributed to the collapse cannot change in this case, it seems the answer is 'no'. Yet, Fig. 6B notes that the vertical lines (the left of which pertains to TP2eco) delimit regions where transient collapse can occur. Please clarify.

- Why don't the TP2eco values correspond between panels a and b, as they do for TP2ecoevo?

- I'm wondering: would a line that bisects the unimodal curves in the per capita growth rate isoclines delimit the optimal fixed trait values across stress levels? If so, I would include this too.

122: 2nd sentence: "and" should be "or"

Fig 2: change "being" to "go"

153: I would help the reader by setting this section off with another subheading that clarifies that all of the results described therein pertain to the exploration of the effect of varying the rate at which the environmental stress changes (and, correspondingly, I'd add a sub heading for the preceding section that clarifies that this section deals with evolutionary response to stress (which is fixed in magnitude)

276: algae should be alga

304: differ should be differing

Referee: 2

Comments to the Author(s)

See attached file

Author's Response to Decision Letter for (RSPB-2021-0277.R0)

See Appendix B.

RSPB-2021-1192.R0

Review form: Reviewer 1

Recommendation

Accept with minor revision (please list in comments)

Scientific importance: Is the manuscript an original and important contribution to its field?

Excellent

General interest: Is the paper of sufficient general interest?

Excellent

Quality of the paper: Is the overall quality of the paper suitable?

Good

Is the length of the paper justified?

Yes

Should the paper be seen by a specialist statistical reviewer?

No

Do you have any concerns about statistical analyses in this paper? If so, please specify them explicitly in your report.

No

It is a condition of publication that authors make their supporting data, code and materials available - either as supplementary material or hosted in an external repository. Please rate, if applicable, the supporting data on the following criteria.

Is it accessible?

N/A

Is it clear?

N/A

Is it adequate?

N/A

Do you have any ethical concerns with this paper?

No

Comments to the Author

I am impressed with the effort the authors put into this revision and feel that it is a significant improvement. I recommend this ms for publication and look forward to citing it. To this end, I just have a few very minor suggested revisions:

lines 324-325: I suggest revising this sentence for clarity to something like: "Different from these studies, here we identified an eco-evolutionary feedback as a mechanism that can underlie such sensitivity."

line 349: I think you want to replace "the level of environmental stress does not change" with "an unchanged environmental stress level"

Decision letter (RSPB-2021-1192.R0)

22-Jun-2021

Dear Ms Chaparro Pedraza

I am pleased to inform you that your manuscript RSPB-2021-1192 entitled "Fast environmental change and eco-evolutionary feedbacks can drive regime shifts in ecosystems before tipping points are crossed" has been accepted for publication in Proceedings B.

The referee(s) have recommended publication, but also suggest some minor revisions to your manuscript. Therefore, I invite you to respond to the referee(s)' comments and revise your manuscript. Because the schedule for publication is very tight, it is a condition of publication that you submit the revised version of your manuscript within 7 days. If you do not think you will be able to meet this date please let us know.

- DNA sequences: Genbank accessions F234391-F234402

- Phylogenetic data: TreeBASE accession number S9123
- Final DNA sequence assembly uploaded as online supplemental material
- Climate data and MaxEnt input files: Dryad doi:10.5521/dryad.12311

[http://datadryad.org/submit?journalID=RSPB&manu=\(Document not available\)](http://datadryad.org/submit?journalID=RSPB&manu=(Document%20not%20available)) which will take you to your unique entry in the Dryad repository. If you have already submitted your data to dryad you can make any necessary revisions to your dataset by following the above link. Please see <https://royalsociety.org/journals/ethics-policies/data-sharing-mining/> for more details.

Sincerely,
Dr Locke Rowe
mailto:proceedingsb@royalsociety.org

Associate Editor
Comments to Author:

Dear colleague
as you will, see the reviewer is very positive about your revised manuscript and has only made a couple of minor suggestions for improvements, which you might want to take into consideration when submitting the final version of the manuscript.

Reviewer(s)' Comments to Author:
Referee: 1

Comments to the Author(s).
I am impressed with the effort the authors put into this revision and feel that it is a significant improvement. I recommend this ms for publication and look forward to citing it. To this end, I just have a few very minor suggested revisions:

lines 324-325: I suggest revising this sentence for clarity to something like: "Different from these studies, here we identified an eco-evolutionary feedback as a mechanism that can underlie such sensitivity."

line 349: I think you want to replace "the level of environmental stress does not change" with "an unchanged environmental stress level"

Author's Response to Decision Letter for (RSPB-2021-1192.R0)

See Appendix C.

Decision letter (RSPB-2021-1192.R1)

25-Jun-2021

Dear Ms Chaparro Pedraza

I am pleased to inform you that your manuscript entitled "Fast environmental change and evolutionary feedbacks can drive regime shifts in ecosystems before tipping points are crossed" has been accepted for publication in Proceedings B.

If you are likely to be away from e-mail contact please let us know. Due to rapid publication and an extremely tight schedule, if comments are not received, we may publish the paper as it stands. If you have any queries regarding the production of your final article or the publication date please contact procb_proofs@royalsociety.org

Data Accessibility section

Open Access

Paper charges

Sincerely,

Appendix A

Review of “Fast environmental change and eco-evolutionary feedbacks can drive regime shifts in ecosystems before tipping points are reached”.

This paper addresses the interesting issue of how evolutionary dynamics can alter transitions in ecological systems between alternative stable states. The authors examine this question with three familiar models that can exhibit alternative states: a single population with an Allee effect; a predator attacking a stage-structure prey population; and, shallow lakes that can alternative between clear and turbid states. They pay most attention to the first and ecologically simplest model. They introduce evolution following standard protocols, where species' traits can evolve according to quantitative genetic rules, with a rate parameter reflecting the availability of standing genetic variation that can fuel natural selection. They demonstrate that allowing evolution can permit persistence of a species that would otherwise collapse, and in general, systems can remain in their original stable state up to higher levels of stressors imposed upon them. They emphasize that it is not just the magnitude of environmental change, but its rate. This is I think a sensible conclusion. They caution in the Discussion that other effects might be observed in complex communities, where many species (such as antagonists⁰ are evolving simultaneously.

The results are interesting, but the paper really needs to be connected better to the existing literature in several respects. This is a missed opportunity. In particular, the authors should relate what they are doing with the Allee effect model to the theme of evolutionary rescue, which is where selection acts rapidly enough to alter the traits of a species in a changed environment, where that species would otherwise go extinct. The first paper using this term was Gomulkiewicz and Holt in *Evolution* (1995), and the topic has received increasing attention in the last several years (see Bell in AREES 2017 for a recent review; there are at least 100 papers since then on this theme). They should also note papers by Andrew Kanarek and others, having to do with how evolution in Allee thresholds can itself permit persistence in novel environments. The authors should make these links in their introduction and in the discussion. In their predator-prey model, the authors only allow the predator to evolve. Prey tend to be more abundant than their predators and often have shorter generation lengths, so might in practice be more likely to evolve, than are predators. The authors should examine what happens when prey evolve, not just the predator.

The really novel result here is sketched on lines 311-319. The authors should expand this and explain what is happening in more detail for each of their three model systems, laying out the causal drivers of this interesting finding.

Appendix B

Swiss Federal Institute of Aquatic Science and Technology
EAWAG
Überlandstrasse 133
CH-8600 Dübendorf

Thursday May 27th, 2021

To the Editor of *Proceedings of the Royal Society B*

Dear Dr Locke Rowe,

Thank you for your email enclosing your comments and reviewers' comments. I greatly appreciate the comments and suggestions. I have carefully reviewed the comments and have revised the manuscript accordingly. You will encounter the responses below in a point-by-point basis. We hope that you find the responses satisfactory and the amended manuscript is now suitable for publication in *Proceedings of the Royal Society B*.

Sincerely,

Catalina Chaparro Pedraza

Associate Editor

Board Member: 1

Comments to Author:

Dear Dr Pedraza

as you will see from the detailed feedback provided, both reviewers are very positive about your work and agree that it would ultimately merit publication. However, both stress that more needs to be done to place your study and results in the context of the existing literature, including the recently published work by Gil et al. in PNAS (2020).

Reviewer(s)' Comments to Author:

Referee: 1

Comments to the Author(s)

In this paper, the authors construct general theoretical models to examine potential interactions among ecological dynamics, rates of phenotypic evolution, and rates of environmental change across three contexts: a population with an Allee effect, a predator-prey system, and an shallow lake ecosystem. Across these contexts, they find that rates of environmental change can, themselves, determine the state of the system, with faster rates of change capable of causing state shifts that are precluded at slower paces of change. They further find that the pace of evolution of organismal phenotypes that dictate adaptability to environmental conditions can counteract this effect, with faster paces of evolution allowing systems to avoid state shifts by essentially keeping pace with environmental change. This work provides new evidence for the importance of both phenotypic diversity and eco-evolutionary feedbacks to both our knowledge and sustainable management of natural ecological systems.

I found this to be an interesting and well written paper with novel insights that will appeal to a broad range of evolutionary biologists and ecologists. This work contributes important evolutionary extensions to recent findings of rate dependence in ecological regime shifts. My main issue with the paper, as written, is that it fails to appropriately acknowledge recent discoveries and, thus, to place the current findings in the context of the literature. Two of the major findings discussed — that evolutionary feedbacks can qualitatively determine whether regime shifts will occur in ecological systems subject to different rates of environmental change, and that, in such cases, regime shifts can occur even when there is one globally stable equilibrium — are novel, and, alone, make this an important contribution. The rest of the major findings — that the state of ecological systems can be sensitive, qualitatively, to the pace and not just magnitude of environmental change and that standard equilibrium analyses can, therefore, mislead — were recently reported in a paper that the authors cite, but without reference to this fact.

We have carefully revised the discussion to appropriately acknowledge that the last two findings were already reported (line 315-325).

For example, the opening paragraph the Discussion, summarizing the major contribution

of this work, fails to acknowledge, and actually contradicts the fact that there is existing ecological theory - cited later in the Discussion - (Gil et al. PNAS 2020) that does posit that stability analysis is insufficient to predict critical thresholds at which regime shifts occur when feedback loops influence transient dynamics. This previous study looked specifically at the interaction between ecological dynamics and rates of environmental change. They did not look at the additional component of differing rates of phenotypic evolution, which is probed in the current manuscript. This is the point of novelty of this manuscript, which corroborates and builds off of the major findings of Gil et al. The Discussion should be revised accordingly, including careful qualification of what specifically sets the current ms apart from this other highly related recent work.

We have rewritten the opening paragraph, and the mention of the lack of ecological theory on the topic of rate-induced regime shifts has been removed. In the revised version of the manuscript, we state what sets this study apart from the study of Gil et al. as suggested by the reviewer (lines 324-325).

Furthermore, the conclusions discussed on lines 353-364 are precisely the same findings revealed by in Gil et al. 2020 (with the exception of the sentence on lines 356-358, which is, indeed, a wholly novel finding). Accordingly, this paper should be clearly acknowledged to clarify that these findings align with those of past work (rather than implying that these are entirely novel findings, as the present presentation does). I suggest carefully reading that paper and revising accordingly.

We have rewritten this paragraph (lines 344-355). In the new version, the focus is on the novel finding mentioned by the reviewer. The rest of the text is moved to the second paragraph of the discussion section where it is discussed in the context of the findings of Gil et al. 2020.

The paragraph that follows (beginning on Lines 365-366) would also benefit from better clarifying the state of the field: Gil et al. 2020 showed that rate-induced regime shifts are possible not in a general model, non system specific model (as in this ms) but in a model parameterized based on data from a real ecosystem: a tropical coral reef. Thus, importantly, Gil et al. has gone a step further than general theory and exemplifies the potential for the kind of rate-dependence emerging from separations in timescales (shown in the current ms) to be an important feature of natural ecosystems.

This paragraph was changed. The new version acknowledges that Gil et al. 2020 show that rate-induced regime shifts can occurred in a model whose parameters were estimated from data (lines 369-371), as suggested by the reviewer.

More specific issues/revisions (fig or line number noted, where appropriate):
Fig. 2: In lower panel, I was struck by the nonlinearity in trait change over env stress - why the acceleration? Is this qualitative shape sensitive to the value of τ ?

The bottom panel in figure 2 shows the mean trait value in the equilibrium as a function of fixed levels of environmental stress. The mean trait value is in equilibrium (i.e. well-adapted to the fixed level of environmental stress) when the fitness gradient equals zero, according to standard quantitative genetics¹. If the mortality is constant and independent of the trait value, then the fitness gradient is affected only by the birth rate. In such case, it is straightforward to infer from the definition of the birth rate (line 47 in supporting information) that the trait value in the equilibrium should be equal to the level of environmental stress. However, the fitness gradient is also affected by mortality due to the trade-off between fecundity and mortality, and this effect increases quadratically with trait value (line 59 in supporting information). Therefore, this causes the nonlinearity mentioned by the reviewer. The parameter τ does not alter the shape of the mean trait value in the equilibrium as a function of fixed levels of environmental stress. The assumptions that led to this nonlinearity are specific of the Allee effect study case. In the other study cases (community and ecosystem level), the model assumptions are very different and the results stated in the section "General patterns in eco-evolutionary systems with ASSs" hold for all of them. Therefore, our general findings are not sensitive to specific model assumptions.

Fig. 3: Is panel A from a model with evolution?

Yes, the results shown in figure 3 correspond to a population subjected to an Allee effect that can evolve. This has been clarified in the figure legend.

I assume so, since that threshold lines up with the bifurcation. If so, this suggests Fig. 2 corresponds to a case when env stress was ramped up from zero quite slowly (over $>10,000$ days), but it's unclear if/how env stress ramping was implemented in Fig. 2. If there was no ramping and, instead, a fixed value was used for env stress, wouldn't that correspond to a much lower threshold (i.e., an immediate increase to the max level would be the fastest possible)? Please clarify.

Figure 2 shows the equilibrium as a function of fixed levels of environmental stress for both systems: the system with only ecological dynamics (blue) and with eco-evolutionary dynamics (red). It does not account for dynamics of environmental stress. A clarification was added in lines 96-98 and in the figure legend.

In addition, I would clarify in the beginning of caption that this is an "evolving population".

This clarification has been included in the caption of figure 3.

Also, it's a bit odd that the trait keeps evolving after the population has collapsed. This makes me wonder: presumably, a small amount of the population persisting, if the trait continues to evolve after the max env change is reached, ought to recover, right? Did you check to see if this was possible? In other words, what prevents the population from

bouncing back, as trait values continually improve over time in fig 3, like fig 5? Please clarify in the main text.

The standard quantitative genetics equation states that the rate of change of the mean phenotype equals the product of the genetic variance and fitness gradient¹. Since genetic variance is assumed constant and larger than 0 (when evolution is enabled), the trait keeps evolving until the fitness gradient is zero. Because the fitness gradient does not equal zero immediately after the population collapses, the mean phenotype keeps changing according to the standard quantitative genetics equation. We agree with the reviewer that it seems odd that the mean trait keeps changing in a collapsed (extinct) population. Therefore, we have added a condition to equation A1.8 (supporting information) to enable evolution only when the population has a positive abundance. As a consequence, figure 3 has changed and now the mean trait does not change after the population collapses. In the other study cases, this condition was also introduced (eq A1.13 and A1.17).

In figure 3 the population does not bounce back like in figure 5 because the system moves from one alternative stable state to another. Notice that in figure 2 the region in between TPc_{eco} and TPc_{ecoevo} is a region of bistability, and the bouncing back (without change in environmental stress) that occurs in figure 5 is possible because the system has only one stable state. A clarification has been added in lines 140-141.

Fig 4C:

I was struck that this effect was only ~18% between thresholds of vastly different trait values. It would be helpful to know what range of effect sizes you observe across parameter values and ecological contexts explored.

Figure 4C shows the effect of mean trait on the location of the tipping point that mark the population collapse. This plot specifically shows this effect for the parameter values for which simulations were performed in figure 3 and 4. Nonetheless, we derived the analytical expression of the function corresponding to this effect in the appendix C. We have expanded the analytical investigation by studying the behavior of this function in appendix C. This result is more general than an exploration of this effect across parameter values and it shows that the tipping point that mark the population collapse increases with increasing mean trait value. We therefore include this analytical result in the main text (eq. 1 lines 153-157).

Additionally, the absolute value of the effect, in this case 18% vs a vast difference in trait values mentioned by the reviewer is relative. We have set as the base line the case in which environmental stress is zero, which does not indicate that the environmental condition is zero, but it indicates the variation in an environmental condition since the base line. A sentence was introduced to clarify it (lines 115-118). In the same way, to facilitate comparison among the ecological only scenario and the eco-evolutionary scenario, we have set the two scenarios to be equal (have same population density and trait value in the equilibrium) in the base line (env stress=0). As with the environmental condition, we defined a value of zero to the base line, which means that we normalized the mean trait value by the trait value at which the population is well-adapted to the

base line. Therefore, the effect that the reviewer mentioned cannot be calculated from the values observed in the figure because they do not correspond to absolute values of an environmental variable neither of a trait.

Fig. 5: I'd recommend aligning the x axes of panels a and b.

Thank you for the recommendation; we have done so.

Also, I would clarify at the start of the caption that this is under the assumption of different rates of environmental change, and I would clarify which rate of change (e.g., in time steps to reach Emax) these solutions pertain to.

The figure 5A shows the population density in the equilibrium as a function of fixed levels of environmental stress. A clarification in the figure legend was added.

5A: are we looking at different rates of env. stressor increase between the evo and ecoevo solutions? Panel B suggests yes, since the TP1eco threshold seems to correspond to a very fast rate of stressor change, while TP1ecoeco seems to pertain to a very slow rate of stressor change. Please clarify.

While figure 5A shows the population density in the ecological (blue) and eco-evolutionary (red) equilibrium as a function of fixed levels of environmental stress, figure 5B shows the occurrence of regime shifts in an evolving population. The word "evolving" was introduced in the legend to clarify. In addition, the parameter listed in the legend shows that $\sigma^2=0.05$ in panel B, indicating that the evolution is enabled.

Fig. 6:

I found Figure 6B challenging to unpack, since it combines many elements in a single visual. Part of what makes this confusing is that thresholds for both the eco and ecoevo models are shown on a 3D plot for which trait values appear fixed (non-evolving). I recommend providing further details in the main text that step the reader through the panel (or breaking it down into simpler sub components).

We have removed the lines indicating the thresholds of TPc_eco and the TPi_ecoevo. In addition, we have rewritten parts of the text to help guide the reader through the figure panels as suggested by the reviewer (lines 193-198).

Additional points of clarification should also address the following:

- It appears that the numerical solutions for the ecoevo feedbacks model presented in Fig. 6A pertain to a trait value of ~ 1.9 , according to panel B. Does this mean, under the specified parameterization, that the ecoevo model evolved to achieve this level of the trait (which is being shown as a fixed trait value in panel B)? I would clarify and, in any case, I think it would be helpful to point out this link to help highlight how the different panels correspond to one another.

In the previous version of the manuscript, figure 6A was showing the effect that the mean trait has on the location of the tipping point that marks the invasion threshold. Now, we have combined panels A and B to make clearer the correspondence between panels as suggested by the reviewer. In the combined plot (now figure 6A), it is clear that the TPi_{eco} (before $TP2_{eco}$) corresponds to the maximum value of environmental stress at which a positive per capita growth rate is possible, and that happens when the trait value is 2.01. The TPi_{eco} is the (invasion) tipping point of the eco-evolutionary dynamics that was identified in the stability analysis (figure 5A). The link between this plot and the bistability analysis has been included in the legend of the figure 6.

- Can a transient population collapse occur for the model with no evolution? Since the trait value that contributed to the collapse cannot change in this case, it seems the answer is 'no'. Yet, Fig. 6B notes that the vertical lines (the left of which pertains to $TP2_{eco}$) delimit regions where transient collapse can occur. Please clarify.

Indeed, the answer is no and a clarification was added (lines 189-192). The line in the left in the previous version of figure 6B did not correspond to $TP2_{eco}$ (now TPi_{eco}) but to $TP1_{eco}$ (now TPc_{eco}) – btw we have changed the notation of the tipping points with numbers to c (collapse) and i (invasion) to avoid confusion. TPc_{eco} and TPi_{eco} delimit the region where the transient collapse can occur, as indicated now by the striped region in figure 5B.

- Why don't the $TP2_{eco}$ values correspond between panels a and b, as they do for $TP2_{eco}$?

In the previous version of figure 6, $TP2_{eco}$ was not indicated in panel A and B. Instead, $TP2_{eco}$ and $TP2_{eco}$ were indicated in the panel A, whereas $TP1_{eco}$ and $TP2_{eco}$ (limits of the region where the transient collapse occurs) were indicated. We realized the use of numbers can be confusing, therefore we have replaced 1 and 2 by c (collapse) and i (invasion), respectively in the new manuscript version.

- I'm wondering: would a line that bisects the unimodal curves in the per capita growth rate isoclines delimit the optimal fixed trait values across stress levels? If so, I would include this too.

The message that we intend to communicate in figure 6 is that the invasion capacity of the population depends on both the level of environmental stress and the trait value. We consider that the information regarding the optimal trait values do not contribute to convey this message and can distract attention. Furthermore, taking into account the opinion of the reviewer who mentioned that figure 6B, now modified into 6A, is "challenging to unpack", adding this information can lead to confusion.

122: 2nd sentence: "and" should be "or"

The change was done.

Fig 2: change "being" to "go"

The change was done.

153: I would help the reader by setting this section off with another subheading that clarifies that all of the results described therein pertain to the exploration of the effect of varying the rate at which the environmental stress changes (and, correspondingly, I'd add a sub heading for the preceding section that clarifies that this section deals with evolutionary response to stress (which is fixed in magnitude)

Thank you for the suggestion, the subheadings in these two sections are indeed very helpful and were added.

276: algae should be alga

The change was done.

304: differ should be differing

The change was done.

Referee: 2

Review of "Fast environmental change and eco-evolutionary feedbacks can drive regime shifts in ecosystems before tipping points are reached".

This paper addresses the interesting issue of how evolutionary dynamics can alter transitions in ecological systems between alternative stable states. The authors examine this question with three familiar models that can exhibit alternative states: a single population with an Allee effect; a predator attacking a stage-structure prey population; and, shallow lakes that can alternative between clear and turbid states. They pay most attention to the first and ecologically simplest model. They introduce evolution following standard protocols, where species' traits can evolve according to quantitative genetic rules, with a rate parameter reflecting the availability of standing genetic variation that can fuel natural selection. They demonstrate that allowing evolution can permit persistence of a species that would otherwise collapse, and in general, systems can remain in their original stable state up to higher levels of stressors imposed upon them. They emphasize that it is not just the magnitude of environmental change, but its rate. This is I think a sensible conclusion. They caution in the Discussion that other effects might be observed in complex communities, where many species (such as antagonists) are evolving simultaneously.

The results are interesting, but the paper really needs to be connected better to the existing literature in several respects. This is a missed opportunity. In particular, the authors should

relate what they are doing with the Allee effect model to the theme of evolutionary rescue, which is where selection acts rapidly enough to alter the traits of a species in a changed environment, where that species would otherwise go extinct. The first paper using this term was Gomulkiewicz and Holt in *Evolution* (1995), and the topic has received increasing attention in the last several years (see Bell in AREES 2017 for a recent review; there are at least 100 papers since then on this theme). They should also note papers by Andrew Kanarek and others, having to do with how evolution in Allee thresholds can itself permit persistence in novel environments. The authors should make these links in their introduction and in the discussion.

We thank the reviewer for suggesting to connect our results to evolutionary rescue, this helps to place our study in the context of existing literature. We added in the text the link in the introduction (lines 45-46) as well as in the discussion (lines 356-366) including citations to the relevant studies.

In their predator-prey model, the authors only allow the predator to evolve. Prey tend to be more abundant than their predators and often have shorter generation lengths, so might in practice be more likely to evolve, than are predators. The authors should examine what happens when prey evolve, not just the predator.

The main findings of our study are 1) that eco-evolutionary feedbacks make ecosystems sensitive to the rate of environmental change, determining whether regime shifts will occur in ecological systems subject to different rates of environmental change, and 2) that such regime shifts can occur even when there is one globally stable equilibrium. Our goal was to demonstrate the generality of these findings across different ecological systems, however, for the sake of simplicity, we limit our investigation to the case in which the most abundant species in the ecological state at low levels of environmental stress evolves, as indicated in the discussion (line 381). This choice enabled us to observe the same asymptotic behavior across systems (that evolution shifts the tipping point to higher environmental stress level), facilitating its comparison. As mentioned in the discussion, we expect that the evolution of the antagonistic species shifts the tipping point in the opposite direction. With some examples shifting the tipping points to higher and others to lower environmental change, the comparison of the asymptotic behavior across systems may be more challenging, but in the end, the temporal dynamics of interest (rate-induced regime shifts) would also be found because it only depends on the existence of eco-evolutionary feedbacks, regardless of what species is evolving. We therefore consider that including the analysis suggested by the reviewer, namely examining what happens when the prey evolves, would not alter the two main findings of our study but would make more challenging this study for the readers.

Furthermore, although we agree with the reviewer that preys might be more likely to evolve in general due to their shorter generation times, the case in which only the predator evolves has been very relevant in literature (e.g. ²⁻⁴). In particular, in the case that motivated the analysis of this model (lines 214-221), top predators are often the focus of research due to their economic interest in fisheries^{3,4}. We therefore believe that the analysis of the case when only the predator evolves is justified for the scope of this study and is relevant according to existing literature.

Nonetheless, understanding how multiple coevolving species (e.g. prey and predator or macrophyte and algae) affect the dynamics of ecological systems under stress should be investigated, as suggested by the reviewer. However, given the complexity of the dynamics already found in this study, we plan to extend these models to include coevolving dynamics in our future research.

The really novel result here is sketched on lines 311-319. The authors should expand this and explain what is happening in more detail for each of their three model systems, laying out the causal drivers of this interesting finding.

The lines mentioned by the reviewer correspond to the section “General patterns in eco-evolutionary systems with ASSs under stress”, which is a summary of our findings across the three model systems as mentioned at the end of the introduction. These findings and their causal drivers are explained in detail in the analysis of each of the models. The details can be found accordingly:

- The statements in lines 311-314 (now 293-296) are explained in detail in lines 122-145 for the Allee effect model, in lines 320-326 (in supplementary material) for the predator-prey model, and in lines 350-356 (in supplementary material) for the shallow lake model.
- The statements in lines 314-318 (now 296-300) are explained in detail in lines 166-203 for the Allee effect model, in lines 329-336 (in supplementary material) for the predator-prey model, and in lines 359-365 (in supplementary material) for the shallow lake model.
- The statement in lines 318-319 (now 300-301) is explained in detail in lines 146-161 for the Allee effect model, in lines 326-328 (in supplementary material) for the predator-prey model, and in lines 356-358 (in supplementary material) for the shallow lake model.

References

1. Lande, R. Natural Selection and Random Genetic Drift in Phenotypic Evolution. *Evolution (N. Y.)*. **30**, 314–334 (1976).
2. Cattau, C. E., Fletcher, R. J., Kimball, R. T., Miller, C. W. & Kitchens, W. M. Rapid morphological change of a top predator with the invasion of a novel prey. *Nat. Ecol. Evol.* **2**, 108–115 (2018).
3. Shackell, N. L., Frank, K. T., Fisher, J. A. D., Petrie, B. & Leggett, W. C. Decline in top predator body size and changing climate alter trophic structure in an oceanic ecosystem. *Proc. R. Soc. B Biol. Sci.* **277**, 1353–1360 (2010).
4. De Roos, A. M., Boukal, D. S. & Persson, L. Evolutionary regime shifts in age and size at maturation of exploited fish stocks. *Proc. R. Soc. B Biol. Sci.* **273**, 1873–1880 (2006).

Appendix C

Referee: 1

Comments to the Author(s).

I am impressed with the effort the authors put into this revision and feel that it is a significant improvement. I recommend this ms for publication and look forward to citing it. To this end, I just have a few very minor suggested revisions:

lines 324-325: I suggest revising this sentence for clarity to something like: "Different from these studies, here we identified an eco-evolutionary feedback as a mechanism that can underlie such sensitivity."

I have made the change suggested by the reviewer

line 349: I think you want to replace "the level of environmental stress does not change" with "an unchanged environmental stress level"

I have changed the sentence to improve readability.

In addition to the changes suggested by the reviewer, I have replaced the pronouns corresponding to first plural person by first singular person. This is because the paper has a single author, however, to avoid bias in the revision process I used the plural.